# The cost of sleeping sickness vector control in Yasa Bonga, a health district in the Democratic Republic of the Congo

Rian Snijders[1,2,3]*, Alexandra P. M. Shaw[4,5], Richard Selby[6], Inaki Tirados[7], Paul R. Bessell[8], Alain Fukinsia[9], Erick Miaka[9], Fabrizio Tediosi[2,3], Epco Hasker[1], Marina Antillon[2,3]

1 Institute of Tropical Medicine Antwerp, Antwerp, Belgium, 2 Swiss Tropical and Public Health Institute, Basel, Switzerland, 3 University of Basel, Basel, Switzerland, 4 University of Edinburgh, Edinburgh, United Kingdom, 5 AP Consultants, Walworth Enterprise Centre, Andover, United Kingdom, 6 Sightsavers, Haywards Heath, West Sussex, United Kingdom, 7 Liverpool School of Tropical Medicine, Liverpool, United Kingdom, 8 Independent consultant, Edinburgh, United Kingdom, 9 Programme National de Lutte Contre la Trypanosomiase Humaine Africaine, Kinshasa, Democratic Republic of the Congo

* riansnijders@hotmail.com

**Data Availability Statement:** All relevant data are within the paper and its Supporting Information files.

## Abstract

Gambiense human African trypanosomiasis (gHAT), a neglected tropical disease caused by a parasite transmitted by tsetse flies, once inflicted over 30,000 annual cases and resulted in an estimated half a million deaths in the late twentieth century. An international gHAT control program has reduced cases to under 1,000 annually, encouraging the World Health Organization to target the elimination of gHAT transmission by 2030. This requires adopting innovative disease control approaches in foci where transmission persists. Since the last decade, case detection and treatment, the mainstay of controlling the disease, is supplemented by vector control using Tiny Targets, small insecticide-treated screens, which attract and kill tsetse. The advantages of Tiny Targets lie in their relatively low cost, easy deployment, and effectiveness. The Democratic Republic of Congo (DRC), bearing 65% of the 799 gHAT cases reported globally in 2022, introduced Tiny Targets in 2015. This study estimates the annual cost of vector control using Tiny Targets in the health district of Yasa Bonga in the DRC and identifies the main cost drivers. Economic and financial costs, collected from the provider's perspective, were used to estimate the average cost of tsetse control expressed as cost (i) per target used, (ii) per target deployed, (iii) per linear kilometre of river controlled, and (iv) per square kilometre protected by vector control. Sensitivity analyses were conducted on key parameters for results robustness.

The estimated annual economic cost for protecting an area of 1,925 km$^2$ was 120,000 USD. This translates to 5.30 USD per target used each year, 11 USD per target deployed in the field, 573 USD per linear km treated, and 62 USD per km$^2$ protected. These costs in the DRC are comparable to those in other countries. The study provides valuable information for practitioners and policymakers making rational, evidence-based decisions to control gHAT.

**Funding:** This study was funded by the Bill & Melinda Gates Foundation (grant OPP1155293) to MB and EH and the Margaret A. Cargill Foundation within the framework of 2 projects aiming to eliminate HAT in two health districts in the Democratic Republic of Congo to EH and Prof. Dr. Marleen Boelaert (PI). Human African Trypanosomiasis Modelling and Economic Predictions for Policy (HAT MEPP) project [OPP1177824 and INV- 005121] supported MA and FT. The funders had no role in the study design, data collection, analysis, or article publication.

## Author summary

In the fight against Gambiense human African trypanosomiasis (gHAT), a devastating disease transmitted by tsetse flies, significant progress has been made through international efforts. Despite the annual cases being reduced to under 1,000, the World Health Organization aims to eliminate gHAT transmission by 2030. A key component of this strategy involves innovative approaches, such as the use of Tiny Targets–small, cost-effective, insecticide-treated screens that attract and kill tsetse flies. This study focuses on the Democratic Republic of Congo (DRC), which bears a substantial burden of gHAT cases, estimating the annual cost of vector control using Tiny Targets in the Yasa Bonga health district. The analysis, conducted from the provider's perspective, reveals an annual economic cost of 120,000 USD for protecting a 1,925 km$^2$ area. This translates to 5.30 USD per target used, 11 USD per target deployed, 573 USD per linear km treated, and 62 USD per km$^2$ protected. These findings, comparable to costs in other countries, offer valuable insights for practitioners and policymakers, guiding evidence-based decisions on cost-effective strategies for gHAT control.

## Introduction

Sleeping sickness or Human African Trypanosomiasis (HAT) is a vector-borne parasitic disease that caused several major outbreaks in sub-Saharan Africa, killing thousands of people during the last epidemic of the 1990s. The disease is transmitted through the bite of an infected tsetse fly (genus *Glossina*) and is almost always lethal if left untreated [1]. This article focuses on the cost of vector control, one of the approaches to control the disease, by reducing the population of tsetse flies responsible for transmitting the parasite [2].

By 1960, colonial authorities almost eliminated HAT, previously a major public health problem affecting millions of people in Sub-Saharan Africa resulting in over half a million deaths between 1940 and 1960. This decrease was achieved by implementing–occasionally oppressive–case-finding measures and providing effective but highly toxic treatments. Unfortunately, the disease re-emerged and peaked at the end of the 1990s, with over 30,000 new cases reported annually, causing a significant social and economic impact on the affected regions. Stepping up HAT control and surveillance efforts from the late 1990s onwards reversed this epidemiological trend, with around 6,200 cases reported in 2013 [3,4].

That year, the World Health Assembly endorsed the goal of HAT elimination in light of the sustained decrease in the disease burden, a better understanding of the disease's epidemiology, and the prospect of improved diagnostics and treatment regimens that were less toxic than previously. The World Health Organization (WHO) set 2020 as the target date for HAT elimination as a public health problem, defined as reducing HAT incidence to fewer than 1 new case per 10,000 population in at least 90% of foci and to fewer than 2,000 cases reported globally. They also targeted 2030 as the year for disease elimination, defined as zero disease incidence [2,4–6]. In 2020, the target of elimination as a public health problem was largely achieved, with only 565 new HAT cases reported globally, 70% of which were identified in the Democratic Republic of Congo (DRC) [7]. In 2022, 516 or 65% of the 799 gHAT cases reported globally were detected in the DRC [3].

Two forms of HAT exist in humans, caused by two subspecies of the parasite *Trypanosoma brucei*, namely *T. b. gambiense* infections currently responsible for over 85% of all HAT cases reported worldwide and *T. b. rhodesiense*. Both forms of HAT are targeted for elimination as a

public health problem, but only *gambiense* HAT (gHAT) is targeted for elimination of transmission to humans as it is presumed to be an anthroponotic infection, unlike *T. b. rhodesiense* HAT, which can infect animals [7].

gHAT is the only form of HAT present in the DRC, and HAT control in this context focuses on clearing the parasite from humans [8]. This strategy is based on a multi-faceted approach, focusing mainly on early case detection and treatment [2]. HAT diagnosis is difficult because of its non-specific symptoms, the diagnostic algorithm's complexity, and the disease's focal distribution [1]. Therefore, an exhaustive screening strategy, even with innovative diagnostics and treatment, requires major investments in equipment, diagnostics, and human resources [2,9]. Even though case-finding strategies have proven to be effective, there are still several foci where transmission persists, sometimes even after HAT case detection and management has been maintained for many years, most likely due to insufficient coverage of the population at risk, limited sensitivity of diagnostic methods and the disappearing awareness and expertise of medical staff [10,11].

In the past, vector control methods, such as vegetation clearing, or insecticide spraying, and in particular, trapping, were occasionally used to control gHAT but were often considered ineffective, too expensive, or too complicated to implement in remote, resource-constrained settings [12–15]. Today, a new, more straightforward and low-cost method has been developed to control populations of riverine tsetse that transmit *T. b. gambiense*, namely 'Tiny Targets'. These small, impregnated screens consist of one 25cm by 25cm square blue cloth, flanked by an insecticide-impregnated mesh of the same size, deployed along the banks of rivers and water bodies where tsetse concentrate. Tsetse are attracted by their blue colour and contact the targets, picking up a lethal dose of insecticide. In 2011, Tiny Targets were introduced in Guinea and Uganda where their relative entomological and economic cost as compared to previous methods has been discussed [16–18]. Afterwards, this relatively Tiny Targets were effectively deployed in several other countries (e.g., Chad, Côte d'Ivoire), achieving in all locations a decline in the tsetse population of 60–95% [19–21]. HAT transmission models estimated that at least a 72% reduction in the tsetse population is required to stop transmission and that the 2030 gHAT elimination goal would be achieved by including a moderately effective tsetse control (60% tsetse population reduction) in the overall gHAT control strategy. Therefore, vector control could be crucial in eliminating gHAT cost-effectively [18,22].

In 2015, gHAT vector control using Tiny Targets was implemented at the health district level in the Yasa Bonga in the DRC for the first time. An evaluation of the impact of Tiny Targets on the *Glossina fuscipes quanzensis*, the primary tsetse vector of gHAT in the DRC observed a reduction in fly catches of more than 85% [13]. This study estimates the annual financial and economic cost of Tiny Target deployment in Yasa Bonga and identifies its main cost drivers.

## Materials and methods

### Research setting

The health system of the DRC is organized at different levels, where every province is subdivided into several health districts where a district team manages a network of health centres and a district hospital. Each health district generally covers a human population between 100,000 and 200,000 which according to national standards is subdivided into health areas of around 10,000 inhabitants each, covered by at least one integrated health centre [23].

As of 2014, a first project started in the health districts of Yasa Bonga and Mosango focussing on improving HAT control named "Integrated HAT control, a model district in DR Congo" and by the end of 2015 a second project was introduced in the same districts named

"TRYP-ELIM. A demonstration project combining innovative case detection, tsetse control and IT to eliminate sleeping sickness at district level in the Democratic Republic of Congo". These projects aimed to effectively eliminate HAT transmission within three years from a health district in the DRC through intensified systematic screening and case management of at-risk populations combined with vector control. In the context of this project, tsetse vector control with Tiny Targets (manufactured by Vestergaard, Lausanne, Switzerland) was implemented throughout the HAT-endemic health district of Yasa Bonga in Kwilu province, formerly Bandundu province. Yasa Bonga is a rural health district with, in 2018, a total population of 235,696 people scattered over 305 villages in an area of 2,810 km$^2$ [13,24]. Over 45% of all HAT cases detected in the DRC were reported in the former Bandundu province between 2000 and 2012, with the highest annual incidence, of 40 cases/10,000 population (208 new cases), being reported in Yasa Bonga. [25].

### Vector control with Tiny Targets in the study area

Fig 1 shows the gradual scale up of vector control with Tiny Targets in the Yasa Bonga health district along the riverbanks of some of the three main rivers (Lukula, Kafi, Inzia). In 2015, the rivers highlighted in red were treated, the following year the river to the north was included as well, and in 2017 treatment was extended to include part of the river forming the western border of the health district. Thus, by 2017, the health areas highlighted in green were covered so around 70% of the health district surface was protected.

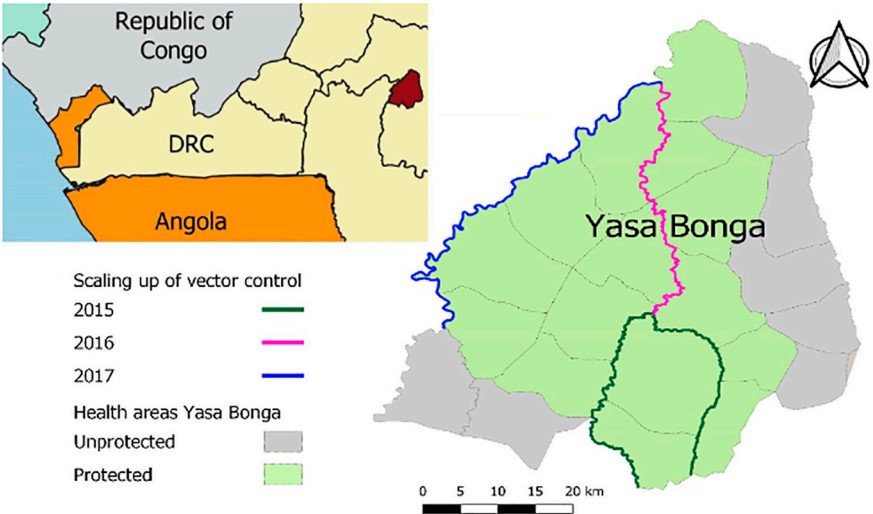

**Fig 1. Map showing the border of Yasa Bonga and the individual health areas.** Overlain is the scaling up of vector control in the health district of Yasa Bonga between 2015 and 2017 Map generated using QGIS 3.28.3 [26,27].

The deployments, as outlined in Table 1, occurred biannually during the dry seasons (July and August and January/early February). To evaluate the impact of the Tiny Targets, the abundance of tsetse flies was assessed before and after deployments. Pyramidal traps were used to compare the number of flies captured during these periods [28]. Teams of day workers without previous experience deployed the Tiny Targets. Local recruitment and training were managed by the Programme National de Lutte contre la Trypanosomiase Humaine Africaine (PNLTHA) and with support from entomologists from the PNLTHA and the Liverpool School of Tropical Medicine (LSTM).

**Table 1. Estimated financial cost of the full scale intervention for a 5-year period based on the costs reported between 2015 and 2017 (USD).**

| Description/cost category | Total Y1 | Total Y2 | Total Y3 | Total Y4 | Total Y5 | Total Y1-Y5 |
|---|---|---|---|---|---|---|
| Monitoring/Surveillance | 14,214 | 14,214 | 14,214 | 14,214 | 14,214 | 71,071 |
| Sensitization | 44,217 | 19,541 | 27,717 | 19,541 | 27,717 | 138,731 |
| Biannual target deployment | 57,571 | 57,571 | 57,571 | 57,571 | 57,571 | 287,855 |
| Management support costs | 52,476 | 13,305 | 13,780 | 13,755 | 19,286 | 112,601 |
| **Total cost** | **168,478** | **104,630** | **113,282** | **105,080** | **118,788** | **610,257** |
| Cost Tiny targets (1$) | 22,622 | 22,622 | 22,622 | 22,622 | 22,622 | 113,110 |
| **Total cost with Tiny target donation** | **145,856** | **82,008** | **82,008** | **82,458** | **96,166** | **497,147** |

Teams assembled targets on-site by gluing the fabric on locally sourced wooden supports. Using traditional canoes or "pirogues", the teams travelled down the river deploying the targets aiming for 50-meter intervals on both riverbanks preferring shorter intervals when exact 50m distances were impossible due to unsuitable deployment or canoe docking sites, explaining a higher actual number of targets deployed than the targeted 40 targets/linear km. The last deployment, covering the whole protected area, involved around 54 targets / linear km, resulting in 11,311 targets over 210 linear km, as illustrated in S1.I Annex. GPS points were recorded for every target placed. Once a routine was established, the teams typically covered around 40 linear km of river weekly (i.e., 40km x 40 targets/km = 1600 targets/week) though this distance increased over the years. A detailed description of the vector control intervention and the impact on the vector, can be found in Tirados *et al.* [13]. Table II in S1 Annex shows the calendar with the different vector control activities during the study period including the number of targets deployed and linear km treated per deployment during the scale up.

Awareness-building activities accompanied the vector control intervention to inform villages about sleeping sickness and tsetse control activities. A vector control management unit was set up at the provincial level after the study period in 2019, which manages a standardized comprehensive vector control sensitization strategy in all villages where vector control measures are implemented. In Yasa Bonga, the vector control sensitization campaign targets 165 villages near river banks where human populations are exposed to Tiny Targets and related deployment activities. The strategy involves training the village leader and two community health care workers from each village to do awareness-building in their communities. The trainings occur every two years, with evaluation meetings held every alternate year. These trained individuals conduct at least one day of awareness-building in their communities, using tools such as T-shirts, megaphones, and sensitization picture boxes in the local language, Kikongo, while a banner is installed in each village to reinforce the message. Additionally, radio spots are broadcast in the health district annually six times a day for a month around the period of deployment to further disseminate information. This approach served as a reference for the cost of awareness-building activities in the region. During the study period tsetse vector control was only implemented in Yasa Bonga but in the meantime, a vector management control unit was set up at national level covering 11 health districts in 2 provinces. These management costs were not taken into account in the main analysis but their cost impact was assessed in the sensitivity analysis.

## Cost methodology

The cost analysis adopted a provider's perspective focusing on costs incurred to implement HAT vector control using Tiny Targets by the Ministry of Health's national elimination

program, namely the PNLTHA. The study only considered costs related directly to the implementation of vector control and omitted research costs as well as costs of geostatistical modeling before the intervention to predict tsetse habitat distribution [29,30]. A full costing approach was adopted, in line with similar analyses in other countries, as similar programs are assumed to be implemented anew in different locations [17,31,32].

The study collected financial and economic costs at local prices between January 2015 and September 2017 from routine activities, expense reports, budgets, and discussions with experts. During this period, five Tiny Target deployments took place, covering a gradually expanding area, as shown in Fig 1 with the linear km treated and targets and traps used detailed in Table II in S1 Annex.

These financial costs represent the monetary expenditures by PNLTHA directly associated with the implementation of vector control. They encompass the tangible, measurable, and explicit financial outlays required for the implementation of the activity. In contrast, economic costs go beyond the explicit financial expenses and consider opportunity costs—the value of resources that could have been used elsewhere but were allocated to vector control and the value of unpaid inputs such as donated drugs or targets or unpaid local community labour [33]. In this provider's perspective study, as PNLTHA paid for all staff, vehicles and other inputs into the control activity, there were few differences between the financial to the economic viewpoint. For example, in this study both staff and vehicles were employed full-time on the VC work, and none were shared with other activities or were borrowed from other organisations, as was the case in some of the recent HAT vector control activities in other countries [31,32]. However, some of the management activities benefitted objectives outside of gHAT vector control.

The study considered resources with a useful life of less than 12 months as recurrent costs and resources with a useful life of over a year as capital costs [34]. For the economic analysis, the capital costs were annualized using straight line depreciation and assuming they had no residual value at the end of this period. Thus their purchase value was divided by their lifespan. Useful life estimates were based on discussions with experts and on WHO-Choice guidelines [35].

A mixed-methods approach was used to estimate the annual costs to treat 210 linear km and to protect 1,925 km$^2$, the extent of the whole operation after 5 deployments (Fig 1 and Table II in S1 Annex). Although PNLTHA was the sole implementer, costs were recorded in different locations and at different levels of the organization. Thus, bottom-up micro-costing was used to estimate costs that were directly allocated to target deployment in the field. We collected detailed data on the quantities of inputs and prices to value the resources used. Additionally, we used a step-down or gross costing approach to estimate costs that could not directly be attributed to specific field activities, such as management and transport costs [36]. The current awareness-building and management strategy were costed using information from current activities (2018–2019), as a standardized strategy for these activities was only implemented after the study period. The management support unit at the provincial level oversaw the vector control activities in four health districts, including Yasa Bonga. Therefore, in the economic analysis, the annual cost of management support was divided by four to calculate the management costs attributable to the health district of Yasa Bonga.

A 5-year period was selected for estimating the financial costs for treating the whole area, with the aim of showing what the provider's costs over time would be for funding such an operation. The time period chosen reflected that capital equipment with the highest cost, namely the vehicles, which were determined to have an estimated lifespan of 5 years according to discussions with field experts. Thus, we took the costs collected for the actual

operation and compared them to the estimated cost of a 5-year project. Then, we made the appropriate adjustments for converting financial costs to economic costs as explained above, replacing capital costs with annual depreciation and adjusting the share of management costs. These costs were not discounted as we only looked at the cost of one year of vector control deployment.

Afterwards, the results were combined to calculate the total annual cost to implement tsetse control covering the entire health district and the cost per activity, namely entomological monitoring and surveillance, sensitization, biannual target deployment, and management support costs. The costs were presented in four main categories: human resources (HR), transport-related costs (fuel, vehicle use and maintenance, and rent of pirogue), specialized equipment, and other (stationary, small camping equipment, etc.) [17,31,32]. Then we looked at the main cost drivers, namely specific expenses that significantly influenced or contributed to the overall costs.

In order to be able to compare the results from Yasa Bonga with gHAT vector control using Tiny Targets in other settings, the costs were also expressed as a cost per (i) area protected (USD/ km$^2$), (ii) cost per target used (USD/target), (iii) target deployed (USD/target), (iv) length of river controlled (USD/km), and person protected (USD/person). The Tiny Targets are deployed along rivers, assuming that each intervention protects a 5-kilometer zone on either side of the river. This means each river is surrounded by a protected corridor that is 10 kilometres wide in total when targets are deployed on both sides. In regions with closely spaced rivers, their 5-kilometer protection zones can overlap. Consequently, the actual protected area can be less than the sum of the individual zones, as overlaps must be accounted for to avoid double-counting. Conversely, people living outside the 5-kilometer zone are also protected when they travel towards the rivers to farm, fish, or for other activities so that the area effectively protected can extend beyond this zone. Targets deployed refers to the number of targets set up in the area at a specific time, typically during a deployment cycle, which constitutes a snapshot measure that reflects how many targets are physically in place to protect the area at any given time. In contrast, the number of targets used annually accounts for all the targets utilized throughout the year. Since targets are redeployed biannually, the annual target count will be higher, roughly double the targets deployed. Tsetse flies along the main rivers are identified as the primary source of infection at the health district level.

We performed a univariate sensitivity analysis for the economic cost of the main cost drivers to evaluate the impact of these drivers on the overall costs. This analysis included considering annual and triannual sensitization, varying the transport costs (+- 10%), including the cost of the vector control unit at central level and varying the number of health districts management by a vector control unit at provincial level. Lastly, the results were compared with cost estimates of vector control with Tiny Targets in other settings.

All costs were recorded in the currency in which they were incurred and converted to USD using the average exchange rate between January 2015 and September 2017, which were Euro to dollar: 1.13 and Congolese franc (CDF) to dollar: 0.00084.

This study, focused on the cost analysis of tsetse vector control for sleeping sickness prevention in the Democratic Republic of the Congo, and did not involve live participants. As such, the data collection and study objectives did not require interactions with human subjects, and therefore, an Ethics Statement is not applicable to the nature of this research.

## Results

Over a 5-year period, the projected annual financial costs for covering the whole area protected ranged from 104,630 USD to 168,478 USD, with the highest cost in the first year

reflecting the initial capital investments required, as illustrated in Table 1. Sensitization and management support costs were highest in the first year, while costs associated with monitoring, surveillance, and target deployment remained consistent across the years. The number of targets deployed increased between 2015 and 2017, correlating with the expanded linear coverage, as illustrated in Fig 1. For a detailed breakdown of yearly costs during the scale-up, refer to Table III in S1 Annex. The overall financial cost over a 5-year period is projected to decrease by approximately 113,000 USD or almost 20% if the targets are donated by the manufacturer.

As illustrated in Table 2, the total average annual economic cost for vector control in the health district of Yasa Bonga to treat 210 linear km is 120,127 USD. Almost 50% of the cost is linked to the biannual target deployment, and the management and sensitization each represent around 20% of the costs (see Fig 2). The cost per target used was estimated at 5.31 USD, 573 USD per linear km of river treated, or 62.40 USD per km$^2$ protected. Costs that could not be directly attributed to a specific category were reported under "Other," such as glue to assemble the targets, camping equipment, phone, and internet credit. How financial costs were converted into economic costs is shown in S2 Annex and Tables IV.1 to Tables IV.4 in S1 Annex also details the cost per deployment and monitoring and surveillance round.

**Table 2. Estimated average annual economic cost to cover 210 linear km based on the costs reported between 2015 and 2017 (USD).** Calculations are shown in S2 Annex.

| Activity/Cost category | Total USD | % |
|---|---|---|
| **Monitoring/Surveillance** | **14,214** | **12** |
| HR | 2,610 | 2 |
| Transport | 5,443 | 5 |
| Traps | 4,189 | 3 |
| Other | 1,972 | 2 |
| **Sensitization** | **26,929** | **22** |
| HR | 2,888 | 2 |
| Transport | - | 0 |
| Specialized equipment | 15,964 | 13 |
| Other | 8,077 | 7 |
| **Biannual target deployment** | **57,571** | **48** |
| Tiny Targets | 26,399 | 22 |
| HR | 14,016 | 12 |
| Transport | 10,198 | 8 |
| Other | 6,959 | 6 |
| **Management support costs** | **21,413** | **18** |
| HR | 10,345 | 9 |
| Transport | 9,035 | 8 |
| Specialized equipment | - | 0 |
| Other | 657 | 1 |
| Trainings/meetings | 1,317 | 1 |
| **Total cost** | **120,127** | **100** |
| **Cost per target used (22,622)** | **5.31** | |
| **Cost per target deployed (11,311)** | **10.62** | |
| **Cost per linear km (209.5)** | **573** | |
| **Cost per km$^2$ protected (1,925)** | **62.40** | |

The main cost drivers were the purchase and import of Tiny Targets and traps (29%), human resources during the deployment and monitoring (16%), sensitization equipment (15%) and fuel (12%) (Fig 3). This is also reflected in the breakdown of the costs per activity as shown in Fig 2.

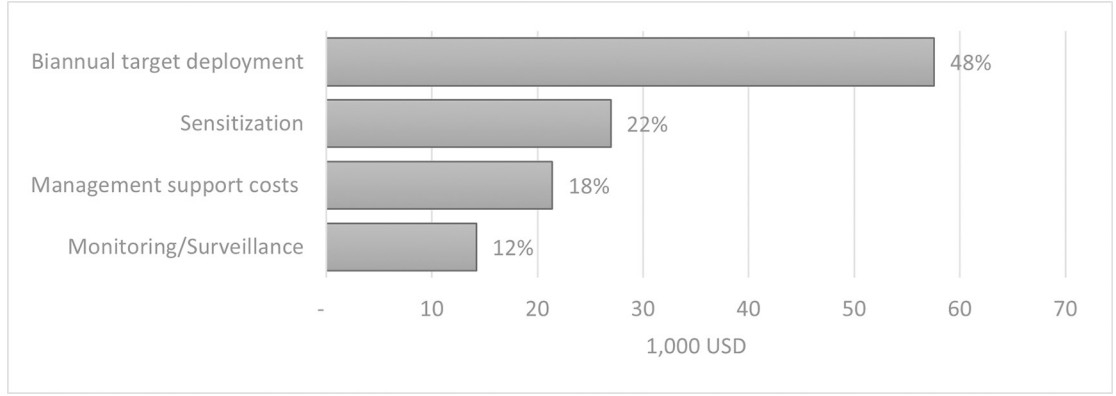

**Fig 2. Contribution of the different activities to the overall annual economic cost (in 1,000 USD).**

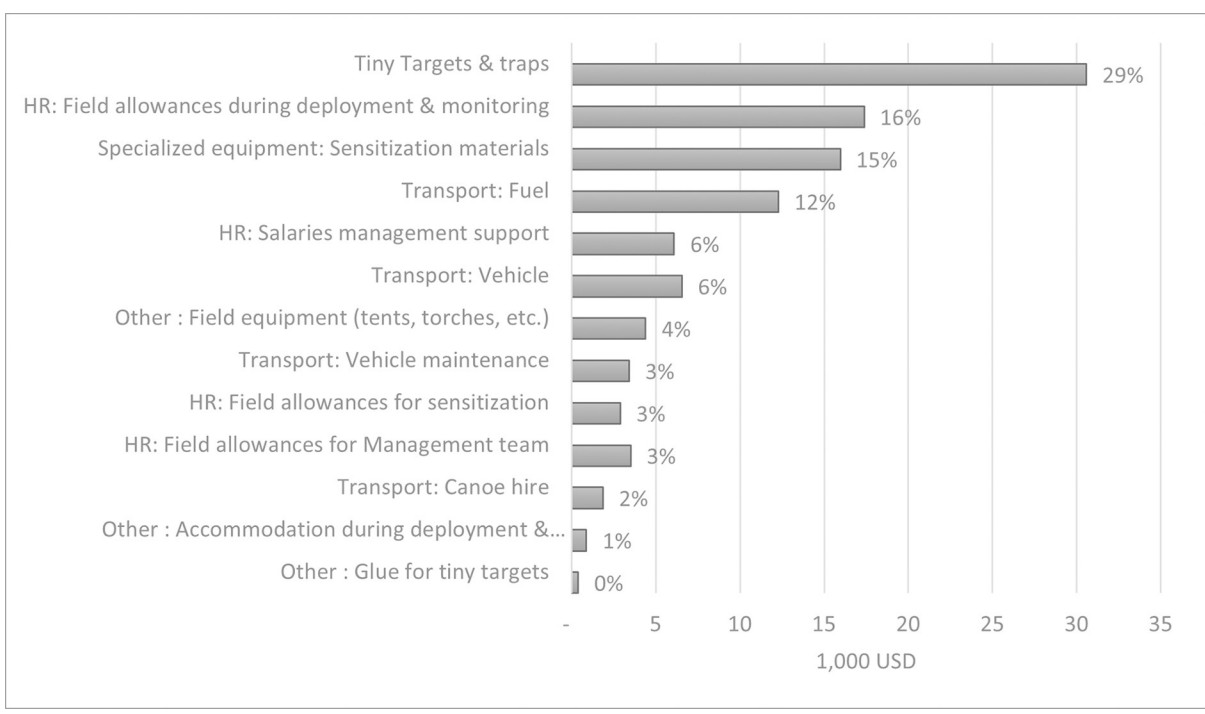

**Fig 3. Contribution of the main cost drivers in relation to the different categories (in 1,000 USD).**

## Sensitivity analysis

The sensitivity analysis looked at changes in the main cost drivers on the economic cost per $km^2$ and the economic cost per linear km of vector control deployed (Figs 4 and 5). Changes

in the transport cost had a minor impact on the overall cost. The cost would decrease by 10% if the sensitisation would take place once every three years instead of biannually. The most significant impact on the cost can be seen in the coverage of the management unit, namely the number of health districts the current set-up of the management unit could accompany and supervise. Setting up such a unit for a limited number of health districts would drastically increase the overall cost of vector control. On the other hand, the cost would decrease if health districts could independently deploy vector control with limited strategic support from the provincial level; provincial resources could be used to cover a larger area provided that this does not negatively impact the quality and traceability of the intervention. If a provincial unit were able to cover 20 health districts the overall cost would decrease by 55%.

Furthermore, the estimated cost did not take into account the cost of the vector control management unit at central level, which supports 11 health districts in 2 provinces. Including this cost would increase the overall cost by approximately 6%.

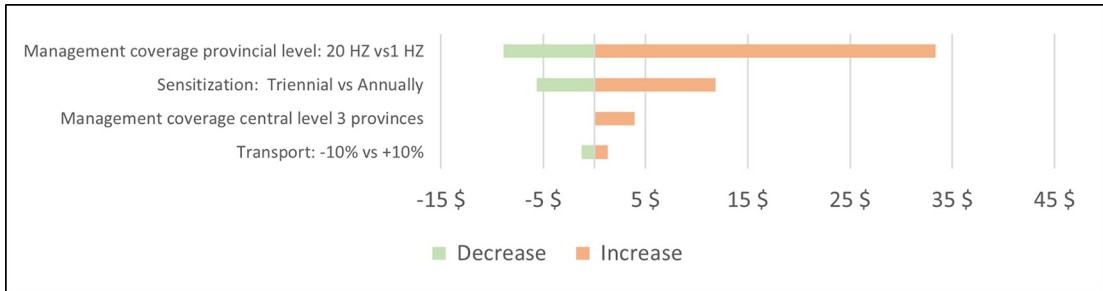

**Fig 4. Sensitivity analysis–Economic cost per km² protected.**

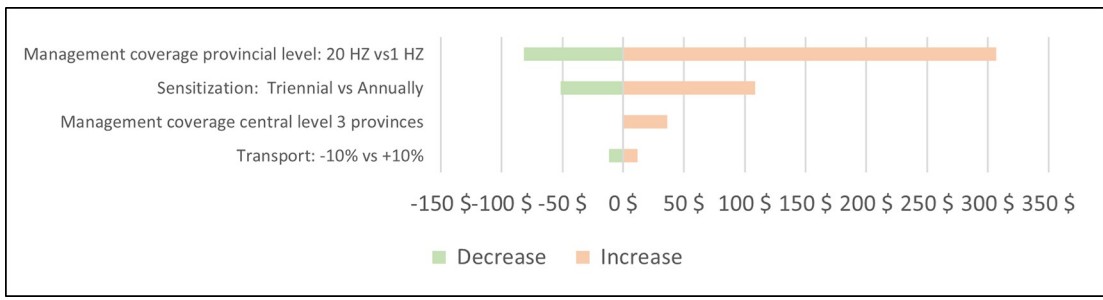

**Fig 5. Sensitivity analysis–Economic cost per linear km.**

## Comparison of cost of HAT vector control using Tiny Targets in other settings

Table 3 compares the cost of gHAT vector control using Tiny Targets in Yasa Bonga, with those in Mandoul (Chad), Bonon (Côte d'Ivoire), and Arua (Uganda). While Tiny Targets effectively reduced the tsetse fly population across all regions, the intervention costs varied significantly depending on the metrics used, driven by the geographical, ecological, and operational contexts unique to each region.

**Table 3. Comparison of annual vector control activities and costs in different settings.**

| Description (costs in USD) | Mandoul, Chad 2015–2016 [21,31,32] | Bonon, Côte D'Ivoire 2016–2017[1] [19,31] | Arua, Uganda 2012–2013[2] [17,18,31] | Yasa Bonga, DRC 2015–2017[3] [13] |
|---|---|---|---|---|
| **Annual economic cost** | **56,133** | **61,253** | **21,982** | **120,127** |
| Number of targets deployed in the area covered | 2,708 | 1,939 | 1,551 | 11,311 |
| Target maintenance | Annual redeployment | Annual redeployment | 60% redeployed in 2012/13 | Biannual redeployment |
| Annual number of targets used in the covered area | 2,708 | 1,939 | 2,501 | 22,622 |
| **Annual cost per target deployed in the area covered** | **21** | **32** | **14** | **11** |
| Number of linear km treated | 45 | NA | 78 | 210 |
| Targets per linear km | 60 | NA | 20 | 54 |
| **Annual cost per linear km treated** | **1,247** | **NA** | **283** | **573** |
| Number of km² treated | 45 | 130 | 16 | **NA** |
| Targets per km² treated | 60 | 15 | 97 | **NA** |
| **Annual cost per km² treated** | **1,247** | **471** | **1,374** | **NA** |
| Number of km² protected | 840 | 130 | 250 | 1,925 |
| Targets per km² protected | 3 | 15 | 6 | 6 |
| **Annual cost per km² protected** | **67** | **471** | **88** | **62** |
| Number of people protected | 39,000 | 120,000 | 112,500 | 165,062 |
| Population density (person/km² protected) | 46 | 923 | 400–500 | 86 |
| **Annual cost per person protected** | **1.44** | **0.51** | **0.20** | **0.73** |

[1] NA–Not applicable as Tiny Targets were deployed throughout the intervention area, rather than along a river.

[2] Costs were converted to 2016 prices [31]. Half way through year 2012–13 for which the costs were analysed, about 60% of targets were replaced. In subsequent years, Uganda moved to biannual redeployment as in DRC. The number of people protected in Uganda was estimated at between 100,000 and 125,000.

[3] NA–Not applicable as Tiny Targets were deployed on the narrow fringing vegetation of the rivers.

In Mandoul, the intervention targeted a localized, inaccessible tsetse habitat within a swamp, requiring a high target density along riverbanks to be effective and prevent people from entering the swamp during deployments. Despite the high personnel costs due to the need for staff from the distant capital, N'Djamena, the total cost per "protected km²" was relatively low because the intervention effectively covered a well-defined, small area, the region's sole source of tsetse flies. Consequently, vector control was considered to effectively cover many high-risk villages, resulting in a protected area larger than the 5-kilometer buffer.

In Bonon, targets were deployed across a degraded yet accessible forested region, requiring an intensive target deployment strategy (two-dimensional instead of one-dimensional alongside river banks) with a high density of targets per km². This resulted in a higher cost per km² due to both the target density and staff costs, despite the relatively small intervention area.

In Arua, the intervention was conducted along narrow riverine forests, primarily focusing on the river systems within the Nile basin. Given the proximity of the villages to these rivers, the treated and protected areas largely coincided, simplifying cost calculations. However, the cost per area protected was influenced by the complexity of the river network and the need to treat multiple rivers. The frequent overlap of their 5-kilometer buffers resulted in a smaller protected area than the number of km treated.

In Yasa Bonga, DRC, the remote, forested location posed significant logistical challenges, requiring intensive deployment strategies to ensure effective coverage. The annual cost per target deployed in Yasa Bonga was lower than in other countries, largely due to the use of locally recruited staff and lower associated supervision costs. The cost per area protected in Yasa

Bonga was similar to that in Arua and Mandoul. However, since Yasa Bonga is more densely populated than Mandoul, the intervention was cheaper regarding the cost per person protected, yielding a value similar to that in Côte d'Ivoire [17,31,32].

## Discussion

This study estimates the costs of vector control for HAT using Tiny Targets in the Yasa Bonga health district in the DRC. At a total economic cost of 120,127 USD or a financial cost ranging between 104,630 USD and 168,478 USD per annum for protecting an area of 1,925 km. The cost per person protected comes to 0.73 USD or 10.62 USD per target deployed. At the beginning of the Tiny Target operation, financial costs are driven by sensitization and management, after which target deployment dominates the costs. In terms of the average annual economic costs, almost 50% are attributed to deployment, and 20% each to management and sensitization. Monitoring and surveillance account for the lowest proportion of costs, both in economic and financial terms. Currently the Tiny Targets are donated by the manufacturer which is projected to reduce the financial cost by almost 20%.

In this paper we have emphasized both the costing methodology and the nature and detail of the information required. Our objective was to help enable such cost analyses of vector control work to be conducted in other settings, not just retrospectively, at the evaluation stage but also before work is undertaken, when first planning an intervention. Thus, Tables III - IV in S1 Annex contain not just details supporting the calculations presented above, but also of the individual components of the deployment, trap monitoring and sensitisation costs. More information is available in the S2 Annex. The basic methodology of collecting financial costs and adjusting these to better reflect the total economic or societal cost has been explored to some extent in the previous papers on the cost of Tiny Target operations to control HAT transmission but here we emphasize and contrast the two approaches [17,31,32].

Assessing the cost-effectiveness of tsetse control is challenging due to the complexity of factors influencing vector control costs and effects. A comparison of results for the DRC with those from other countries suggests significant variations in vector control costs. In HAT endemic foci, adapting vector control strategies to the local context is essential, introducing variability in the overall costs. Numerous factors contribute to this variability, including diverse tsetse habitats such as expansive forests, mangroves, swamps, and narrow riverbanks. Furthermore, the choice of vector control strategy hinges on considerations such as target coverage, deployment methods (ranging from canoes and on-foot approaches to the use of cars and motorbikes), the availability of local manpower and the frequency of target deployment and monitoring. All these factors contribute to the intricacies of the overall cost.

Defining the effectiveness of vector control for sleeping sickness involves evaluating its impact on reducing disease transmission by tsetse flies, primarily with the preventive objective of mitigating and ideally halting HAT transmission. In contrast, other HAT control measures, namely case detection and subsequent treatment, operate reactively by addressing cases post-infection. Vector control efficacy could be measured by the reduction in cases or the number of cases averted. The prevalence of the disease within a protected area will significantly influence the effectiveness, measured in terms of cases averted or Disability-Adjusted Life Years (DALYs) avoided.

Directly comparing the cost-effectiveness of a vector control intervention is challenging, as it is seldom the only control measure implemented and its impact on transmission is difficult to measure. Therefore, the cost of vector control is typically contextualised using metrics like cost per km treated, cost per km$^2$ protected, or cost per population protected (or when feasible, cost per case averted). However, these indicators are influenced by factors external to the vector control intervention, such as baseline prevalence, population density in the protected area,

or the size of the protected area compared to the treated area. Therefore, it is crucial to consider the denominator when interpreting the cost-effectiveness indicators presented throughout the literature. Vector control with Tiny Targets appears to have a lower cost in terms of cost per person protected in Uganda, Côte d'Ivoire, and the DRC as the tool was deployed in an area with a higher population density, resulting in a relatively lower cost per person protected than in Chad [17,31,32].

For sleeping sickness, an optimal strategy would likely integrate case detection initiatives aimed at promptly identifying and treating infected individuals with vector control methods targeting the reduction of tsetse fly transmission. The synergetic impact of these approaches could yield a substantial reduction in sleeping sickness cases. The evaluation of their individual and combined effectiveness and costs is the focus of the Human African Trypanosomiasis Modeling and Economic Predictions for Policy (HAT MEPP) project [37].

While several studies demonstrated that vector control of tsetse flies can play an essential role in HAT elimination, using Tiny Targets presents several limitations and challenges.

Significantly reducing the HAT disease burden through tsetse control relies on its ability to reduce transmission effectively. HAT tends to persist in remote and rural areas with dense vegetation near water sources such as rivers, lakes, and ponds. The lack of comprehensive, accurate geospatial data on tsetse fly habitats and limited information about the actual "transmission zones," the sites where people get bitten by infected tsetse, makes it challenging to identify locations where vector control could impact transmission. This requires geospatial modelling followed by on-site entomological surveys, but limited information is available on the complete actual cost of these preliminary evaluations [17,30–32]. These knowledge gaps also make it challenging to develop a uniform vector control strategy or to determine the necessary "quantity" of vector control per square kilometer needed to halt transmission throughout the countries affected by the disease [19,38].

An exploratory entomological survey in Yasa Bonga conducted during the study period on community-based tsetse control (2017–2018) revealed that fishponds provided suitable habitats for tsetse. Addressing these habitats through large-scale vector control interventions is challenging when conducted by vector control teams unfamiliar with the local environment, as locating and navigating such areas would significantly increases their workload. A study in the southwest of the health district showed that a community-based approach to deploying Tiny Targets organized and managed by local community members could effectively cover these areas. However, data on the costs of this type of intervention is currently unavailable [39]. Additionally, Tirados et al. showed that these habitats in at-risk zones are generally connected to the main river network and are considered part of interconnected tsetse fly populations [13]. Their analysis suggests that targeting the highest-risk areas of the rivers should also impact the associated secondary habitats, making rivers the priority for intervention.

Out of 519 health districts in the DRC, 281 health districts reported one or more cases of sleeping sickness between 2000–2020 [40]. Currently, the DRC vector control for sleeping sickness is successfully being implemented by local teams managed by the PNLTHA. This is taking place in 11 health districts located in two provinces adjacent to the Kinshasa province and accessible by car from the capital. This gives a vector controlled area for current interventions of 10,977 km$^2$. The activities were successfully continued during the COVID-19 pandemic, which shows that a sustainable system was developed that transitioned responsibilities to provincial and health district levels [41]. Introducing vector control activities in remote and resource-constrained areas in other provinces of the DRC will require additional investment for the preliminary geospatial and entomological studies needed, the creation of vector control capacity (resource, equipment, and trained personnel, etc.) and higher transport costs, compared to Bandundu, due to the distances and a higher fuel cost. Initial costs related to training,

infrastructure development, and equipment acquisition would be spread over a shorter duration if the intervention would be implement for a limited period due to the context of disease elimination resulting in a higher average cost. HAT vector control might leverage an existing supply chain and management system by integrating this activity into the broader health system while reinforcing the entire health system beyond this specific disease focus. Antillon et al. conducted a cost-effectiveness analysis of HAT elimination strategies across the DRC [40]. The analysis focused on 165 health districts where active transmission was most likely or had been observed historically. The study considered two alternative strategies for vector control with Tiny Targets, namely full vector control, which involved deploying Tiny Targets along all large rivers in a health district and targeted vector control or the deployment of Tiny Targets along large rivers only in areas with a case density of at least 1 case per 10 km of the treated riverbank. Under the current status quo strategies, which included vector control only in 11 health districts, the probability of achieving end of transmission (EoT) by 2030 is expected to be relatively low, with EoT projected in 117 out of the 165 health districts. The total cost for the status quo strategy from 2024 to 2040 is estimated at $171 million (95% CI: $89 million—$283 million), with $2.1 million attributed to vector control. When strategies are optimized to maximize the probability of achieving EoT by 2030, including full vector control in 45 health districts and targeted vector in 27 districts, the total economic cost from 2024 to 2040 is estimated at $216 million (95% CI: $149 million—$354 million), with $34 million attributed to vector control. This optimized strategy is expected to increase the number of health districts achieving EoT by 2030 to 138 health districts out of 165 [40].

Vector control using Tiny Targets has proven to be a feasible tool at a lower cost than former methods [42]. Therefore, Tiny Targets can play an important role in the HAT elimination strategy as it could help stop transmission in foci where the disease persists. The successful scale-up of a Tiny Targets will require a good local understanding of the terrestrial and aquatic ecosystem of tsetse fly habitats and the development of tsetse control measures where provincial and health administration levels play an important role. The implementation cost of this approach can be drastically reduced when implemented on a large scale, with a local vector control management unit covering a larger geographical area for minimum duration of 5 years, allowing full use of the investments needed to build local capacity and awareness. While any measure aimed at elimination will present some challenges, this study, like other costing studies on Tiny Targets, shows that the cost can be quite accessible.

## Supporting information

**S1 Annex. Supplementary information.** I. Illustrations of vector control, II. Calendar of vector control activities between 2015 and 2017, III. Details Financial costs, IV. Details Economic costs.
(DOCX)

**S2 Annex. Supplementary spreadsheet.**
(XLSX)

## Acknowledgments

The authors acknowledge the staff of the PNLTHA and the fieldworkers in Yasa Bonga for their contributions under arduous field conditions. The authors also would like to thank Professor Stephen Torr and Dr. Andrew Hope for their valuable contributions to the final draft of the article.

## Author Contributions

**Conceptualization:** Rian Snijders, Richard Selby, Inaki Tirados, Alain Fukinsia, Erick Miaka, Epco Hasker.

**Data curation:** Rian Snijders, Richard Selby, Alain Fukinsia.

**Formal analysis:** Rian Snijders, Alexandra P. M. Shaw, Paul R. Bessell.

**Funding acquisition:** Inaki Tirados, Erick Miaka, Epco Hasker.

**Investigation:** Rian Snijders, Alexandra P. M. Shaw, Richard Selby, Inaki Tirados.

**Methodology:** Rian Snijders, Richard Selby, Inaki Tirados, Fabrizio Tediosi, Marina Antillon.

**Project administration:** Rian Snijders, Richard Selby, Inaki Tirados, Alain Fukinsia, Erick Miaka, Epco Hasker.

**Resources:** Rian Snijders.

**Supervision:** Rian Snijders, Richard Selby, Inaki Tirados, Erick Miaka, Fabrizio Tediosi, Epco Hasker, Marina Antillon.

**Validation:** Rian Snijders, Richard Selby, Inaki Tirados, Paul R. Bessell, Alain Fukinsia, Marina Antillon.

**Visualization:** Rian Snijders.

**Writing – original draft:** Rian Snijders.

**Writing – review & editing:** Rian Snijders, Alexandra P. M. Shaw, Richard Selby, Inaki Tirados, Paul R. Bessell, Alain Fukinsia, Erick Miaka, Fabrizio Tediosi, Epco Hasker, Marina Antillon.

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
