## [Decision Letter · Decision Letter 0]

7 Aug 2024

Dear Mrs. Snijders,

Thank you very much for submitting your manuscript "The cost of sleeping sickness vector control in the Democratic Republic of the Congo" for consideration at PLOS Neglected Tropical Diseases. As with all papers reviewed by the journal, your manuscript was reviewed by members of the editorial board and by several independent reviewers. The reviewers appreciated the attention to an important topic. Based on the reviews, we are likely to accept this manuscript for publication, providing that you modify the manuscript according to the review recommendations. 

Sincerely,

Sergio Recuenco

Academic Editor

Nigel Beebe

Section Editor

Reviewer's Responses to Questions

**Key Review Criteria Required for Acceptance?**

**Methods**

-Are the objectives of the study clearly articulated with a clear testable hypothesis stated?

-Is the study design appropriate to address the stated objectives?

-Is the population clearly described and appropriate for the hypothesis being tested?

-Is the sample size sufficient to ensure adequate power to address the hypothesis being tested?

-Were correct statistical analysis used to support conclusions?

-Are there concerns about ethical or regulatory requirements being met?

Reviewer #1: The objectives are very clear exposed and the study is adequate, but in my opinion there is an important mistake as it is not evident that the results obtained from only one district (Yasa Bonga) can be extrapolated to a whole country (DRC). This mistake is easily remediable changing the objective to estimate the costs of tsetse vector control to just to this district. In this sense, the title could be incorrect and should be modified to "the costs of sleeping sickness vector control in Yasa Bonga, DRC". This mistake should be also addressed in the abstract and discussion.

In this sense, the authors clearly state in the discussion (line 426-427). "In HAT endemic foci, adapting vector control strategies to local context is essential, introducing variability in the overall costs". DRC is a vast country with many variabilities between the different regions that make difficult the extrapolation from Yasa Bonga to the whole country.

Reviewer #2: The study presents a clearly articulated objective - costing tiny target deployment. Methods are described in detail. As the data being reported are actual rather than sampled programme costs, statistical considerations such as sample size and hypothesis tests are not required.

**Results**

-Does the analysis presented match the analysis plan?

-Are the results clearly and completely presented?

-Are the figures (Tables, Images) of sufficient quality for clarity?

Reviewer #1: Taking into account the concern referred to the area of study (Yasa Bonga and not DRC), the results are very clear and the data provided in the supplementary files very completed. To include the details in the annexes makes easily readable and understandable the paper.

May be, it could be good to explain more clearly the difference between "targets used" and "targets deployed".

It is not clear how the area treated/covered by tsetse vector control has been calculated. I understand this aspect falls out of the purpose of this paper but it could be interesting to give some elements to know how it has been estimated. In fact. it looks like that there are areas considered as covered by vector control but in some of these areas, the closer Tiny Target is a more than 10 km (estimated from the map provided). I don't know the region but authors refers that fishponds are widely distributed and can play a role in the presence of tsetse flies. Probably, there are also other small streams and tributary rivers on top of the three identified main rivers of the health district. Could they play a role in maintaining tsetse presence in theses areas? Is the same method used in other regions used in the comparison of the cost?

In graph 1 the different color between rivers covered starting 2015 and 2017 is not easily distinguishable. If possible, it is recommended to change the color.

Reviewer #2: Results are clearly presented with appropriate detail.

**Conclusions**

-Are the conclusions supported by the data presented?

-Are the limitations of analysis clearly described?

-Do the authors discuss how these data can be helpful to advance our understanding of the topic under study?

-Is public health relevance addressed?

Reviewer #1: I have to acknowledge that when I received the paper, I was a bit misleaded by the title: I thought it will give an idea about how much is currently invested in tsetse vector control in DRC and how much will be needed to implement tsetse vector control, as recommended in the DRC, but this was not the case.

It is not clearly stated in the paper but it looks like that vector control has been introduced in very limited health districts of the country (11 health districts and 2 provinces). How many health districts are considered endemic for HAT in DRC?. According to the paper used as reference 7, there are 507,000 square km at risk of HAT in DRC (2016-2020) with more than 42 million people living in this area. How many square km are currently covered by tsetse vector control?. Ideally, how many should be covered? According to the data obtained, which cost would have to cover all these areas? These figures will give really an idea of the cost of tsetse vector control in DRC as originally stated in the paper. This could be interesting to give an idea about the huge magnitude of the problem and the financial gap currently existing to reach the target of elimination of transmission, the limitations to extend the vector control activities, and why and how to prioritize vector control in some areas. This aspect is not considered in the discussion neither in the conclusions and it could be of high interest.

Reviewer #2: A careful and insightful discussion is presented which highlights the interpretation and the limitations of the analysis. The discussion would benefit from a small amount of additional reflection on the importance of sensitisation activities to the effectivness of the intervention and the amount of time spent by community health workers supporting the intervention and whether this was or should have been costed.

**Editorial and Data Presentation Modifications?**

Reviewer #1: Title: 

- Specify that the study is in Yasa Bonga and not all the country (DRC).

Abstract: 

- Specify that the study is in Yasa Bonga and not all the country (DRC).

- You refer that the disease resulted in half a million deaths in the late twentieth century. This data is not shown in the introduction of the paper (you just refer thousands of people), so please include it also in the text with the adequate reference.

Introduction:

- Lines 76-81: Please provide a reference for the first statement in the second paragraph ("By 1960,... and economic impact on the affected regions.").

- Lines 85-88: In the first statement of the third paragraph you refer to the "international community". This is quite vague. Please could you specify and provide a reference for this statement.

- Lines 135-139: May be to complete the paragraph, please indicate in how many districts and provinces tsetse vector control is currently used in DRC. Later on (lines 212-213) you refer that vector control management covers "11 health districts in 2 provinces". How many provinces and health districts are at this time considered as endemic for HAT?

- Line 139: Specify the area of study.

Material and methods:

- Line 172-173: I don't understand the conclusion stated as "... so that almost the whole district was protected". Checking the map provided, if I'm not wrong, it looks like 15 health areas were considered as "covered by vector control" but it looks like the district has 22 health areas. So 15/22 areas of the health district were covered by vector control. How do you consider 15/22 as almost the whole health district ?

- Line 209-210: You refer that 165 villages (over 305 villages of the district) were identified for sensitization. May be it could be interesting to give a short explanation about how these villages were identified.

Results:

- Table 3: Note that the comparison is between 4 specific and well defined endemic areas from 4 different countries. It will be more precise to indicate in the title row "Mandoul, Chad" (instead of "Chad"), "Bonon, Côte d'Ivoire" (instead of Côte d'Ivoire"), "Arua, Uganda" (instead of "Arua) and "Yasa Bonga, DRC" (instead of "DRC").

Discussion:

- Line 479: May be better "2017-2018" instead of "'17-'18"

References:

- There is a typo with reference 14, as lines are cut in different paragraphs.

Reviewer #2: see above re sensitisation and community health worker role.

**Summary and General Comments**

Reviewer #1: Interesting paper, easily readable with clear results and discussions. The details in calculations are particularly good to understand the methods used and to do similar exercises. In my opinion, it includes an important mistake as it considers that the data obtained are applicable to DRC, when there were obtained just in one specific health district (Yasa Bonga) and it is no evident that this can be extrapolate to the whole country. I would strongly suggest that the paper is focusing just in this district and later in the discussion consider the value of the data for the whole country. In this sense, a most appropriate title could be "the cost of sleeping sickness vector control in the health district of Yasa Bonga, DRC" and to introduce the adequate changes in the abstract and in the discussion. This should be on line with previous similar articles, referred specifically to Arua and not Uganda (ref 17 and 20), to Bonon and not Côte d'Ivoire (ref 31 and 19), Mandoul and not Chad (ref 32 and 21). I think it is very important to include this consideration in the Title and Abstract of the paper.

Reviewer #2: This study provides useful information for health economists and policy makers looking to understand the likely costs of vector control and what drives these costs. The information is provided in sufficient detail for those in other settings to adjust costs to their contexts. With minor adjustments to the discussion to touch on the importance of sensitisation to the effectiveness of vector control, and the potential burden on community health workers and how this was costed, I am happy to recommend this for publication.

PLOS authors have the option to publish the peer review history of their article (what does this mean?). If published, this will include your full peer review and any attached files.

Reviewer #1: No

Reviewer #2: No

Figure Files:

Data Requirements:

Reproducibility:

References

---

## [Decision Letter · Decision Letter 1]

1 Nov 2024

Dear Mrs. Snijders,

We are pleased to inform you that your manuscript 'The cost of sleeping sickness vector control In Yasa Bonga, a health district in the Democratic Republic of the Congo' has been provisionally accepted for publication in PLOS Neglected Tropical Diseases.

Best regards,

Sergio Recuenco

Academic Editor

Nigel Beebe

Section Editor

Shaden Kamhawi

co-Editor-in-Chief

Paul Brindley

co-Editor-in-Chief

Reviewer's Responses to Questions

**Key Review Criteria Required for Acceptance?**

**Methods**

-Are the objectives of the study clearly articulated with a clear testable hypothesis stated?

-Is the study design appropriate to address the stated objectives?

-Is the population clearly described and appropriate for the hypothesis being tested?

-Is the sample size sufficient to ensure adequate power to address the hypothesis being tested?

-Were correct statistical analysis used to support conclusions?

-Are there concerns about ethical or regulatory requirements being met?

Reviewer #2: (No Response)

**Results**

-Does the analysis presented match the analysis plan?

-Are the results clearly and completely presented?

-Are the figures (Tables, Images) of sufficient quality for clarity?

Reviewer #2: (No Response)

**Conclusions**

-Are the conclusions supported by the data presented?

-Are the limitations of analysis clearly described?

-Do the authors discuss how these data can be helpful to advance our understanding of the topic under study?

-Is public health relevance addressed?

Reviewer #2: (No Response)

**Editorial and Data Presentation Modifications?**

Reviewer #2: (No Response)

**Summary and General Comments**

Reviewer #2: I am satisfied with the response to my feedback and happy to recommend this for publication.

PLOS authors have the option to publish the peer review history of their article (what does this mean?). If published, this will include your full peer review and any attached files.

Reviewer #2: No

---

## [Editor Report · Acceptance letter]

12 Nov 2024

Dear Mrs. Snijders,

We are delighted to inform you that your manuscript, "The cost of sleeping sickness vector control In Yasa Bonga, a health district in the Democratic Republic of the Congo," has been formally accepted for publication in PLOS Neglected Tropical Diseases.

Best regards,

Shaden Kamhawi

co-Editor-in-Chief

Paul Brindley

co-Editor-in-Chief
